# AutoML Algorithms for Online Generalized Additive Model Selection: Application to Electricity Demand Forecasting

Keshav Das[1]  Julie Keisler[1,2]  Margaux Brégère[1,3]  Amaury Durand[1]

[1]EDF R&D, Lab Paris-Saclay
[2]Inria Paris
[3]LPSM, Sorbonne Université

**Abstract**  Electricity demand forecasting is key to ensuring that supply meets demand lest the grid would blackout. Reliable short-term forecasts may be obtained by combining a Generalized Additive Models (GAM) with a State-Space model , leading to an adaptive (or online) model. A GAM is an over-parameterized linear model defined by a formula and a state-space model involves hyperparameters. Both the formula and adaptation parameters have to be fixed before model training and have a huge impact on the model's predictive performance. We propose to optimize them using automated Machine Learning. For this purpose, we define an efficient modeling of the search space (namely, the space of the GAM formulae and adaptation parameters) as well as mutation and crossover operators on this space and apply an evolutionary algorithm to select the best parameters. We evaluate our method on short-term French electricity demand forecasting which demonstrates the relevance of the approach.

## 1 Introduction

Given the challenges of storing electricity on a large scale, it is essential to forecast electricity demand as accurately as possible to ensure an efficient balance between production and demand, thus maintaining the continuous operation of the electricity system. Hence, power stations can calibrate their output by knowing the required electricity demand at any given moment. Generalized Additive Models (GAM) are highly effective electricity demand forecasting models that have been widely used over the last two decades to forecast short- and mid-term demand (ranging from a day to a couple of years) - see, among others, Pierrot and Goude (2011) and Wood et al. (2015). They model the expected value of the random target variable $Y$ with a sum of potentially nonlinear covariate effects, which we call a formula. The forecasts' accuracy depends entirely on the choice of terms in this sum, namely, for each term (or effect), the covariate or covariates and their relationship (linear, polynomial, etc.) with the response variable. Further, in Section 3, we will detail how such a formula $f$ characterizes a GAM.

GAMs are widely used in the electricity field because they combine the flexibility of fully non-parametric models with the simplicity of multiple regression models. Indeed, they are trained on large historical data sets (several years) and make the underlying assumption that the distribution of the target variable is stationary conditionally to covariates. However, they fail to capture recent changes in electricity demand behavior (sobriety, new electrical appliances, etc.). To overcome this issue, models can be made adaptive in a second phase. Specifically, forecasters use the latest data to dynamically reweight the terms in the GAM sum; see Ba et al. (2012) and Obst et al. (2021) for further details. This procedure is detailed in Section 3. We will see that it depends on some hyperparameter $Q$ which needs to be fixed before the adaptation process. Thus, to create a reliable GAM for electricity demand forecasting, forecasters must set a GAM formula and the hyperparameter for

---

The code is available here: `https://anonymous.4open.science/r/dragam-88D0`

its adaptive version. Then, the model is estimated with the Penalized Iterative Re-Weighted Least Squares (P-IRLS) method from Wood et al. (2015), implemented in the `mgcv` R-package.

Choosing the formula and the adaptive hyperparameter is therefore crucial to obtain a good electricity demand model (i.e. one that is efficient or simple to interpret, etc.). Testing a large number of models (i.e., running the P-IRLS algorithm on many formulae) can be challenging, time-consuming, and computationally expensive. In this work, we propose to automate these choices using Automated Machine Learning (AutoML) algorithms. The latter estimate the best adaptive model $(f^\star, Q^\star)$ in a pool of models $\Omega$ containing all the candidates. To choose among candidates, they need to evaluate the performance of any model $(f, Q) \in \Omega$: this is the costly part because it requires training the model and evaluating its performance on a validation data set using a loss function $\ell : \Omega \to \mathbb{R}$. A search algorithm is used to select the most promising models to be tested. A summary of the AutoML framework as well as an outline of the paper are illustrated in Figure 1.

$$\overset{\text{Search Algorithm (Section 5)}}{(f^\star, Q^\star) \in \underset{(f, Q) \,\in\, \Omega}{\operatorname{argmin}} \; \ell(f, Q)} \overset{}{.} \quad \text{Performance Evaluation (Section 4)}$$

Search Space (Section 3)

Figure 1: Summary of the AutoML framework.

After a literature discussion in Section 2, we define the search space in which the AutoML algorithm will search for its solutions in Section 3. In Section 4, we focus on performance evaluation, while Section 5 discusses the search algorithms we consider. Finally, we demonstrate the relevance of our method on French electricity demand data in Section 6.

## 2 Literature discussion

**AutoML.** Automated Machine Learning is made of two main subproblems: model selection and hyperparameter optimization, which can be tackled jointly with the Combined Algorithm Selection and Hyperparameter (CASH) optimization problem (Feurer et al., 2015). The state of the art in AutoML offers both very general algorithms for selecting models and/or hyperparameters that perform well for any type of machine learning model desired (see, for example Akiba et al. 2019 or Tang et al. 2024), and more specialized algorithms for certain types of specific machine learning models, such as models from the scikit-learn library (Feurer et al., 2022) or neural networks (White et al., 2023). To our knowledge, GAMs have only been the subject of one specific approach, GAGAM (Cus, 2020), which is described below. While generic approaches can be applied to GAMs to a certain extent, they quickly become limited when it comes to capturing the hierarchical structure of formulae.

**Model selection for GAMs.** Given a set of features, along with some training data, the Genetic (Evolutionary) Algorithm for Generalized Additive Models (GAGAM) developed in the R language by Cus (2020) applies a sequence of crossovers and mutations to subsets of a randomly initialized population. After each sequence of evolutions, the models are ranked according to their loss, which is calculated using the input training data. Eventually, the aim is that the algorithm will retain the models with the best formula, constructed using some of the input features and by deciding whether they should enter linearly or not. However, the same type of spline and the same degrees of freedom are applied to all nonlinear features, which restrains the search space. Moreover, bi-variate effects are not supported simultaneously with linear features.

**Dragon.** DRAGON, or DiRected Acyclic Graphs optimizatioN, is an open-source Python package offering tools to conceive Automated Machine Learning frameworks for diverse tasks. The package is based on three main elements: building bricks for search space design, search operators to modify those bricks and search algorithms. Originally developed for Deep Neural Networks optimization, the package is based on the encoding of various computer objects (integers, floats, lists, etc.) and can be easily extended to other optimization tasks. Search operators can be used to create neighbors or mutants of a solution, allowing local search or evolutionary approaches to converge on optimal solutions. The DRAGON package implements four search algorithms based on the search operators: a random search, an asynchronous evolutionary algorithm (Keisler et al., 2024b), Hyperband (Li et al., 2018), and Mutant-UCB (Brégère and Keisler, 2024). In this work, we used DRAGON's various building blocks to encode GAMs. We then adapted the search operators to our representation, allowing us to directly use the search algorithms in the package.

## 3 Search space

Introduced by Hastie and Tibshirani (1986) in 1986, Generalized Additive Models (GAM) are semiparametric models widely used for electricity demand forecasting. Before training a GAM and using it for electricity demand forecasting, the GAM formula - i.e. the equation which links the expectation of the target variable $Y$ with the covariates $X$ - and the hyperparameters for its adaptive version must be defined. We detail these two steps below.

### 3.1 Fixed setting

With $g$ a link function (e.g., identity or logarithmic function), a GAM models the value $g(\mathbb{E}[Y|X])$ as a sum of smooth functions of the covariate $X$. With $K$ the number of terms (also called effects) in the sum, let, for any $k = 1, \ldots, K$, $J_k$ be the set of indices of the covariates associated with the effect $k$ and $f_k$ the function that models the relationship between the target variable $Y$ and the subset $X_{J_k}$ of the covariates $X$; a GAM has a structure like:

$$g\left(\mathbb{E}\left[Y|X\right]\right) = \sum_{k=1}^{K} f_k\left(X_{J_k}\right).$$

As it is classically done for electricity demand forecasting, in all that follows, and without loss of generality, we will consider additive models, namely $g$ is equal to the identity. In our context, the target variable is a time series, and for a time $t$, $Y_t$ is the amount of electricity demanded between $t$ and $t + 1$ (in the experiments of Section 6, the time step is half an hour). The latter depends on various variables $Z_t$, including weather (temperature, wind, etc.) and calendar-related (public holidays, weekdays, etc.) factors. Electricity consumers do not respond instantaneously to changes in the weather, among other things, because of the thermal inertia of buildings. This is why (recent) past weather may influence electricity demand. To be as general as possible, the set of covariates $X_t$ includes present and past covariates, so $X_t = (Z_s)_{s=1,\ldots,t}$. Therefore, we obtain the equation $\mathbb{E}[Y_t|X_t] = \sum_{k=1}^{K} f_k(X_{J_k,t})$. It remains to specify the smooth functions $f_k$ to get a proper formula. This task is divided into two: feature engineering (to extract useful information from $X_{J_k,t}$) and relationship specification (to link this information with the target variable).

**Feature engineering.** For some $k = 1, \ldots, K$, feature engineering for covariates $X_{J_k}$ can provide summarized information and prove extremely valuable. For example, taking into account an exponentially smoothed temperature or selecting a subset of categories from a categorical variable can be achieved by designing parameterized functions denoted by $\psi_k$. These two possibilities are fully detailed in Appendix A. Only the parameters of the feature engineering functions have to be set-up in the formula and these will be the ones optimized by the AutoML algorithm.

---

https://dragon-tutorial.readthedocs.io/en/latest/

**Relationship specification.** For every $k = 1, \ldots, K$, we need a basis of functions in which $f_k$ may be represented. To define the formula, this basis, completely specified by its functions $(b_k^i)_{i=1,\ldots,q_k}$, where $q_k$ is the size of the basis, has to be picked. In our set-up, this means choosing $q_k$ and the type of functions in the basis (linear, cubic spline, tensor product, etc.) - we refer to Appendix A for further details.

Overall, a GAM can be seen as an over-parameterized linear model. The number of effects $K$, and for each $k = 1, \ldots, K$, the subset $J_k$, the function for potential feature engineering $\psi_k$, the size $q_k$ and the spline basis $(b_k^i)_{i=1,\ldots,q_k}$ have to be chosen by the forecaster. It defines the formula $f$ and implicitly assumes that:

$$f(X_t) = \mathbb{E}[Y_t | X_t] = \sum_{k=1}^{K} \sum_{i=1}^{q_k} \beta_{k,i} b_{k,i} (\psi_k(X_{J_k,t})). \tag{1}$$

The coefficients $\beta_{i,k}$ are estimated using the Penalized Iterative Re-Weighted Least Squares (P-IRLS) method from Wood et al. (2015), implemented in the mgcv R-package. This last step can be considered as *model training*, while the choice of the formula corresponds to *model optimization*.

## 3.2 Adaptive setting

GAMs have a good ability to generalize relations between inputs and an output, but they lag behind in terms of adaptability to new data. Hence, adaptive GAMs have been introduced by Ba et al. (2012) to overcome this issue. They are well suited for short-term demand forecasting as they take into account recent data and catch changes in the target variable distribution. With $\mathbf{f}(X_t)$ the $K$-dimensional vector containing all the effects of the estimated GAM $f_k(X_{J_k,t})$, for $k = 1, \ldots, K$; we consider the state-space model approach introduced by Obst et al. (2021) and assume, for $t = 2, \ldots,$

$$\begin{cases} Y_t &= \theta_t^{\mathrm{T}} \mathbf{f}(X_t) + \varepsilon_t, & \text{with} \quad \varepsilon_t \overset{\text{i.i.d}}{\sim} \mathcal{N}(0, \sigma^2) \\ \theta_t &= \theta_{t-1} + \eta_t, & \text{with} \quad \eta_t \overset{\text{i.i.d}}{\sim} \mathcal{N}(\mathbf{0}, \overline{Q}). \end{cases}$$

The variance $\sigma^2$ and covariance $K \times K$ matrix $\overline{Q}$ of the Gaussian white noises $(\varepsilon_t)_t$ and $(\eta_t)_t$, respectively, parameterize the model. This is the setting of Kalman filtering so we may use the recursive formulae of Kalman providing the expectation of $\theta_t$ given the past observations, yielding to a forecast for $Y_t$. As described in Obst et al. (2021), the Kalman filter is crucially dependent on the hyperparameter $Q = \overline{Q}/\sigma^2$. It also requires some initialization parameters that we fix in this work. Thus an adaptive version of a GAM relies on the choice of the parameter $Q$, a $K \times K$-matrix. In all that follows, we consider only diagonal matrices.

Overall, the formula $f$ of a GAM and the parameter $Q$ of the associated state-space model for the adaptive version can be considered as hyperparameters from the standpoint of machine learning; they cannot be modified during training but are significantly tied to a model's accuracy.

## 3.3 Encoding through DRAGON

As mentioned above, we used the DRAGON package to build our search space. DRAGON provides a set of variables for encoding different computer objects. Each element of the search space is represented by a fixed size array containing $f$ and potentially $Q$ in the adaptive setup. The variables $f$ and $Q$ are arrays of variable size, with a dimension $K$ equal to the number of effects in $f$. Each element of $Q$ is a float, while each element of $f$ corresponds to an effect. Each effect is made up of several elements, depending on the type of effect. First, one or two categorical variables corresponding to the features (depending on whether we are in a univariate or bivariate case), from the covariate set $Z$. Depending on the chosen feature(s), components of different types (categorical, floats or integers) can also be added. The variables corresponding to the features and

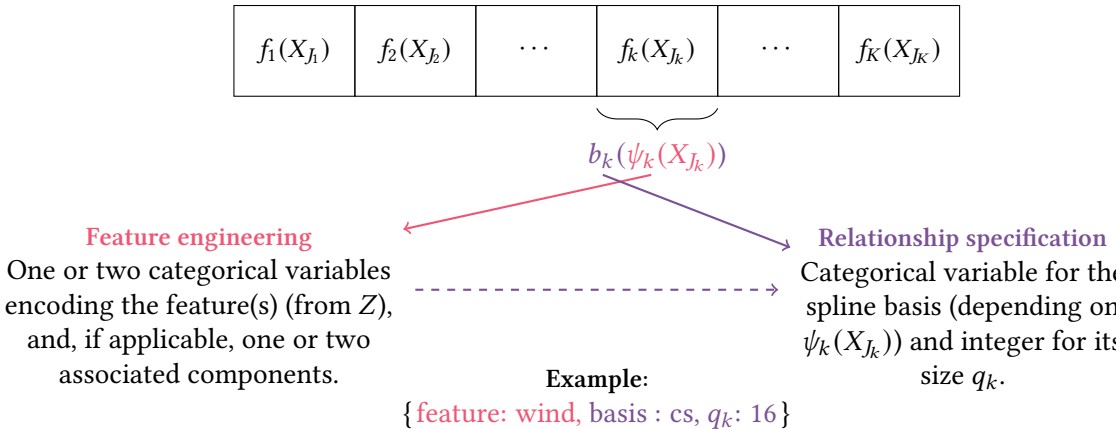

Figure 2: Encoding of a GAM formula $f$ using the DRAGON package.

their components, if any, are used to construct the value $\psi_k(X_{J_k})$ for each time step $t$. Depending on the value of $\psi_k(X_{J_k})$, several spline basis $b_k$ can be used. A categorical variable specifies the one chosen for the effect $k$. Finally, a last attribute $q_k$, an integer, determines the basis size. The search space is hierarchical in the sense that the elements of a given effect $k$ depend on the feature(s) chosen for that effect. Moreover, at the scale of the variable-size array encoding the $f$ formula, the addition of a new effect is done in such a way as to avoid any duplication of a given $\psi_k(X_{J_k})$. An illustration of the encoding of a formula $f$ can be found Figure 2. The process of constructing a GAM formula using DRAGON's tools is fully described in Algorithm 1 in the appendix and works as follows: first, the covariate, or the pair of covariates for bi-variate effects, is selected. Then comes the feature engineering part, and subsequently, the relationship is chosen. Finally, the coefficients $Q$ are also generated. Tuning a GAM with the P-IRLS algorithm can be a long and tedious job. We then aim to apply efficient model selection algorithms to optimize them. We just define the search space $\Omega$ that contains all the possible models we may consider, so basically the pairs of formulae and hyperparameters $(f, Q)$. We now have to define a loss function $\ell : \Omega \to \mathbb{R}$ so that we can compare the models with each other.

## 4  Performance evaluation

We recall that we aim to find the best pair $(f^\star, Q^\star) \in \Omega$ and that this is done by minimizing a loss function $\ell : \Omega \to \mathbb{R}$. For a model $(f, Q) \in \Omega$, we denote by $\widehat{f}$ the fixed generalized additive model obtained by running the P-IRLS algorithm implemented in the mgcv-package on a training data set $\mathcal{D}_{\text{TRAIN}} = (Y_t, X_t)_{t=1,\dots,\tau_1}$. This algorithm estimates the coefficients of Equation (1) by performing a regularized over-parameterized linear regression. The regularization factor, optimized using a cross-validation criterion, avoids over-fitting and ensures that covariate effect functions are smoothed. This criterion involves the effective degrees of freedom of a GAM. It is defined by the trace of the influence matrix: the matrix $A$ such that, for any $t = 1, \dots, \tau_1$, $\widehat{f}(X_t) = AY_t$. We refer to Chapter 4 of Wood (2017) for further details. The effective degrees of freedom of a GAM $\widehat{f}$ may be interpreted as a (possibly non-integer) number of parameters of the model and is thus a good measurement of its complexity; it is denoted by $\mathrm{DF}(\widehat{f})$. For any time step $t$, this fixed model provides the forecast $\tilde{Y}_t = \widehat{f}(X_t) = \sum_{k=1}^{K} \widehat{f}_k(X_{J_k,t})$. To evaluate the quality of the forecasts provided by the model $(f, Q)$ on a validation data set $\mathcal{D}_{\text{VALID}} = (Y_t, X_t)_{t=\tau_1+1,\dots,\tau_2}$ in an adaptive setting, the forecasts are modified. Indeed, for each time step $t$ of the data set, the model provides the forecast $\widehat{Y}_t = \sum_{k=1}^{K} \widehat{\theta}_t \widehat{f}_k(X_{J_k,t})$, where the adaptive vector $\widehat{\theta}_t$ was previously updated using the Kalman filter equations. This calculation involves, among other things, the prediction error $(Y_{t-1} - \widehat{Y}_{t-1})$ and

the hyperparameter $Q$ (for further details, see, e.g. Obst et al., 2021). We consider the Root Mean Squared Error (RMSE) to evaluate the prediction on $\mathcal{D}_{\text{VALID}}$. Maximizing prediction accuracy while avoiding overcomplicated models led us to define, for any model $(f, Q)$, the following loss function:

$$\ell(f, Q, \mathcal{D}_{\text{VALID}}) = \sqrt{\frac{1}{\tau_2 - \tau_1} \sum_{t=\tau_1+1}^{\tau_2} \left(Y_t - \widehat{Y}_t\right)^2} + \eta \text{DF}(\widehat{f}). \tag{2}$$

and $\eta \in \mathbb{R}^+$ is a regularization parameter which balances the two terms of the loss: the RMSE on the validation data set and the degrees of freedom. We emphasize that running the Kalman filter on the whole validation data set is computationally expensive: in our model selection framework, both training and testing are costly.

## 5 Search algorithms

We now focus on the AutoML strategies we consider to solve the minimization problem $\arg\min \ell(f, Q)$ over $\Omega$. It should first be pointed out that once a formula $f$ has been selected, there are several statistical approaches to optimize the choice of the hyperparameter $Q$. For example, in Chapter 5 of de Vilmarest (2022), Algorithm 5 proposes an iterative grid search procedure to tune $Q$. We denote by $Q_{\text{IGS}}(\widehat{f}, \mathcal{D}_{\text{TRAIN}}, Q_0, i)$ the matrix output by this algorithm, after $i$ iterations, for a fixed trained model $\widehat{f}$ and an initial matrix $Q_0$. In the following, we explore algorithms which make use of this iterative grid search optimization and others which view $Q$ as a classical hyperparameter. We used the tools in the DRAGON package to encode the various elements of the search space as detailed in Section 3. We then modified the search operators to use our representations with the package's search algorithms. We first designed a procedure to randomly generate a model $(f, Q)$ and thus create an initial population. It is detailed in Algorithm 1 of Appendix B. Then, we defined specific mutation and crossover operators useful for the Evolutionary Algorithm. In comparison with the search operators that are already present in DRAGON, the ones that have been implemented prevent a covariate from intervening more than once in the formula. This is done to guarantee interpretability. Below, we detail these two processes.

**Crossover**. We used a two-point crossover between two GAMs $(f^1, Q^1)$ and $(f^2, Q^2)$ to create two offsprings $(f^{12}, Q^{12})$ and $(f^{21}, Q^{21})$. It can be done on the formulae of respectively $K^1 < K^2$ terms and eventually on the adaptation parameters as illustrated below. The effects from the formulae and the elements from the matrix are swapped, just like bits in the regular two-points crossover. This is illutrated in Figure 3.

$$
\begin{aligned}
f^1(X) &= f_1^1(X_{J_1^1}) + f_2^1(X_{J_2^1}) + f_3^1(X_{J_3^1}) + \ldots + f_{K_1}^1(X_{J_{K^1}^1}) \qquad & Q^1 &= \text{diag}(q_1^1, \ldots, q_{K^1}^1) \\
f^2(X) &= f_1^2(X_{J_1^2}) + f_2^2(X_{J_2^2}) + f_3^2(X_{J_3^2}) + \ldots + f_{K_2}^2(X_{J_{K^2}^2}) \qquad & Q^2 &= \text{diag}(q_1^2, \ldots, q_{K^2}^2)
\end{aligned}
$$

$$\downarrow \qquad\qquad\qquad\qquad \downarrow$$

$$
\begin{aligned}
f^{12}(X) &= f_1^1(X_{J_1^1}) + f_2^2(X_{J_2^2}) + f_3^2(X_{J_3^2}) + \ldots + f_{K_1}^1(X_{J_{K^1}^1}) \qquad & Q^{12} &= \text{diag}(q_1^1, \ldots, q_{K_1}^2) \\
f^{21}(X) &= f_1^2(X_{J_1^2}) + f_2^1(X_{J_2^1}) + f_3^1(X_{J_3^1}) + \ldots + f_{K_2}^2(X_{J_{K_2}^2}) \qquad & Q^{21} &= \text{diag}(q_1^2, \ldots, q_{K_1}^1, \ldots, q_{K_2}^2)
\end{aligned}
$$

Figure 3: Illustration of the crossover.

**Mutation**. The mutation operator creates a new model from an existing one $(f, Q)$. It may impact a given effect in the GAM formula $f$ by randomly changing the covariate(s), the relationship with the target variable (and, when it is relevant, the number of splines in the basis), or the feature

engineering function. Multiple characteristics can be changed at once. The mutation can also be an effect addition (the effect is then randomly generated) or deletion. We ensure that the new formula is non-empty and that none of the effects appear twice. The coefficient in the matrix $Q$ can be randomly mutated as well. The size of $Q$ matrix is automatically adjusted to the number of terms in the formula.

We are now able to extend the search algorithms from DRAGON to GAM selection. For more information on these algorithms, please refer to Keisler (2025). In the experiments of Section 6, we will consider an evolutionary algorithm (EA). We propose two versions, which are fully described in Algorithm 2 of Appendix B. For the first version, the EA is performed only on the formulae. This first algorithm, referred to as $\mathrm{EA}(f) + \mathrm{Q_{IGS}}$, trains the models with the P-IRLS algorithm so the evolutionary algorithm outputs a fixed model. We finalize the optimization by running the iterative grid search until convergence. For the second one, the adaptive hyperparameter is considered to be part of the search space (so $f$ and $Q$ are optimized at the same time) and we refer to it as $\mathrm{EA}(f, Q)$.

## 6 Experiments: short-term electricity demand

### 6.1 Experiment design

**Data sets and adaptation procedure**. Our experiments consist in forecasting the French hourly electricity demand on a short-term horizon. The open source data set comes from the French Transmission System Operator. The set of covariates contains weather (temperature, cloud cover and wind) and calendar covariates. We work within an operational framework: each week we receive electricity demand data with a delay. Specifically, we assume that every Monday at 8:00 we access new electricity demand records: data from Saturday 00:00 nine days earlier to the previous Friday 23:00. Then, we update our models (namely the vector $\theta_t$ introduced in Section 3.2) using these new data points and we predict the demand for the next seven days (from Monday 8:00 to next Monday 7:00). Models were fitted on the training data set $\mathcal{D}_{\mathrm{TRAIN}}$, which contains electricity demand, weather and calendar information from 2017 to 2020. Since they completely differ from the rest of the data and to avoid some bias on the forecast, we remove the periods of lockdown due to the COVID-19 pandemic from the training data set. The validation data set $\mathcal{D}_{\mathrm{VALID}}$ used to evaluated the models (i.e. to compute the loss defined in Section 4) consists of data from the years 2021 and 2022. Finally, models were tested on $\mathcal{D}_{\mathrm{TEST}}$: the years 2023 and 2024.

**Algorithms**. The performance of the models found by the proposed approach is compared with that of a state-of-the-art handcrafted model detailed in Appendix C and with that of the model found by GAGAM (the R-package developed by Cus, 2020). As the effect of the hour $H_t \in \{0, \dots, 23\}$ is crucial to forecast electricity demand, it is often more efficient to consider a model per hour (see Fan and Hyndman, 2011 and Goude et al., 2013). Thus, we run the algorithms for each hour, independently. The training of the handcrafted model (referred to as "SotA" in what follows) and the one output by GAGAM ("GAGAM") is completed with the optimization of the adaptation parameter using the iterative grid-search algorithm ("$+\mathrm{Q_{IGS}}$"). We executed the proposed AutoML algorithm with a budget of $T = 200$, which means that 200 calls to the P-IRLS algorithm will be made during the entire runtime. All experiments were made with a population of $M = 20$ individuals. Our search space, well designed for our problem, allows us to represent the handcrafted model in order to propose it in the initial population of both EAs. For these search algorithms, the tournament selection size was chosen equal to $k = 5$ (see Algorithm 2 in Appendix).

**Search space**. The search space defined in section 3 is parameterized by several elements, including features, feature modifications, spline bases and the size of these bases. We have chosen these elements based on our industrial knowledge to represent the largest number of GAM formulae that

---

https://www.rte-france.com/eco2mix

we know are relevant to our problem. These choices impact the performance, particularly as we have had to keep the size and complexity of our GAM formulae to a minimum in order to avoid excessive computation time.

**Performance evaluation**. The evaluation of a model $(f, Q)$ starts with its training on $\mathcal{D}_{\text{TRAIN}}$. To get the forecasts $\widehat{Y}_t$ defined in Section 4, the operational adaptation procedure described above involving $Q$ and Kalman filtering is then run. For $EA(f, Q)$, we emphasize that both training and forecasts computation (because of the adaptation) can be time consuming. In contrast, for GAGAM $(+Q_{\text{IGS}})$ and $EA(f) + Q_{\text{IGS}}$, the forecasts are computed in the fixed setting (namely with $\widehat{\theta}_t = 1$ for all $t$ so $\widehat{Y}_t = \widetilde{Y}_t$). Once the forecasts on $\mathcal{D}_{\text{VALID}}$ have been computed, the loss of the model defined in Equation (2) can be obtained. We recall that the loss function balances two terms: the RMSE on $\mathcal{D}_{\text{VALID}}$ and the complexity of the model through the degrees of freedom. To give importance to both terms and find a good trade-off, based on computations of the two terms of several models, we empirically set $\eta = \sqrt{\text{Var}[Y]}/5000$, where the variance is the empirical variance on $\mathcal{D}_{\text{VALID}}$.

## 6.2 Results

We compared the performance of the state-of-the-art handcrafted model ("SotA" and "SotA + $Q_{\text{IGS}}$") and the one found by GAGAM ("GAGAM" and "GAGAM + $Q_{\text{IGS}}$") with the two versions of the evolutionary algorithm (EA): $EA(f) + Q_{\text{IGS}}$ and $EA(f, Q)$. Table 1 presents the RMSE and the MAPE (Mean Absolute Percentage Error) for each data set (training, validation and testing). Figure 4 shows the RMSE for each hour on the testing set. For the fixed setting, GAGAM and $EA(f)$ perform

| Model | Training data set RMSE · MAPE | Validation data set RMSE · MAPE | Testing data set RMSE · MAPE |
|---|---|---|---|
| SotA | 2510 MW · 3.44 % | 2951 MW · 4.33 % | 4397 MW · 7.95 % |
| GAGAM | 2162 MW · 2.82 % | 2776 MW · 4.25 % | 4240 MW · 7.73 % |
| EA($f$) | 1814 MW · 2.33 % | **2440 MW · 3.58 %** | 4290 MW · 8.03 % |
| SotA + $Q_{\text{IGS}}$ | 3301 MW · 4.60 % | 3051 MW · 4.22 % | 3364 MW · 5.07 % |
| GAGAM + $Q_{\text{IGS}}$ | 11208 MW · 10.67 % | 6636 MW · 7.76 % | 5976 MW · 7.78 % |
| EA($f$) + $Q_{\text{IGS}}$ | 1805 MW · 2.32 % | 2466 MW · 3.64 % | 4320 MW · 8.10% |
| EA($f, Q$) | 1594 MW · 2.10 % | **1292 MW · 1.79%** | 1552 MW · 2.20% |

Table 1: RMSE and MAPE on training, validation and testing data.

quite similarly to the SotA model on the testing data set. Errors on the validation data set highlight the efficiency of the evolutionary algorithms and of the proposed search space modeling. For the adaptive setting, GAGAM + $Q_{\text{IGS}}$ and $EA(f)$ + $Q_{\text{IGS}}$ perform much worse than the SotA model. Figure 4 reveals that, for GAGAM + $Q_{\text{IGS}}$, most of this is due to a few hours: GAGAM probably selects over-complicated models. Then, either $Q_{\text{IGS}}$ does not converge to a relevant minimum, or the adaptation process is no longer identifiable at all, leading to bad performances on the three datasets. Comparing the performances of GAGAM + $Q_{\text{IGS}}$ and $EA(f)$ + $Q_{\text{IGS}}$, specifically on Figure 4, legitimates the addition in the loss - see Equation (2) - of the second term: less-complicated models are selected. Finally, it is clear that $EA(f, Q)$ largely outperforms all the previous models and improves the SotA model's performance by 54%. It therefore seems crucial to optimize both the formula and the adaptation parameter simultaneously. These promising results come with new questions and possible improvements. Indeed, with this first work, $f$ and $Q$ are treated by EA as two independent parameters. Doing "smart" mutations and crossing-overs by considering the effect of the GAM $f_k$ "attached" with the associated coefficient in the matrix $Q$ could probably improve the results. Moreover, we believe that $Q_{\text{IGS}}$ (which is time-consuming) could be used partially (a

few iterations) during the AutoML algorithm, and not only at the end. Finally, we highlight an overfitting of our AutoML algorithms on the validation test. We should consider an evaluation process involving methods close to cross-validation.

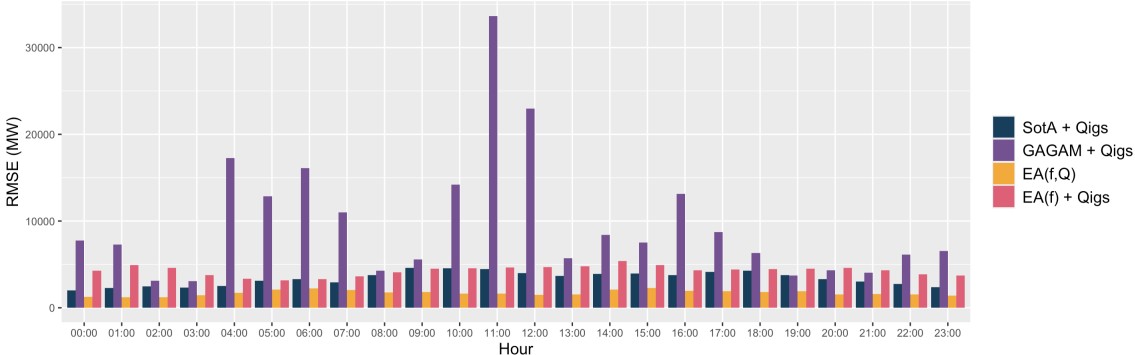

Figure 4: Hourly RMSE on testing data set for the adaptive models SotA + $Q_{IGS}$, GAGAM + $Q_{IGS}$, $EA(f) + Q_{IGS}$ and $EA(f, Q)$.

## 7 Conclusion

This work proposes a framework for automated online generalized additive model selection. It proposes an efficient modeling of the search space which is compatible with the DRAGON framework. Application to short-term electricity demand forecasting attests the relevance of the approach and highlights how it is crucial to optimize both the formula of the GAM and the adaptation parameter simultaneously. In practice, it is possible to greatly improve electricity demand forecasts by using mixtures of models (see for example Gaillard et al. (2016)). The idea is to have several good predictors, called "experts", who make quite different errors from one another, and to make time-evolving weighted averages of their predictions. This AutoML work on GAMs could be improved by trying to optimize not a single formula but a mix of formulae, or even by integrating different types of models into the search space for the best mix. The homogenization, through the DRAGON package, of the representation of the GAMs in this work with that of the deep learning models proposed in EnergyDragon (Keisler et al., 2024a; Keisler and Brégère, 2024), which are also applied to electricity demand forecasting, would make it easy to design search spaces containing both neural networks and GAMs. These two types of models naturally model electricity demand in very different ways and therefore do not make the same errors, which would probably produce a good mix. Finally, these model choices, whether in a mix or not, could vary over time. While online tuning can produce gains, too much change in the data distribution may require online AutoML.

## 8 Broader Impact Statement

Our work automates the creation of reliable load forecasting models. It could provide different actors in the electricity system with access to reliable electricity demand forecasts without the need to spend a lot of time building models by hand. While this work is limited to short-term national forecasts, GAMs can be used for different forecast horizons and spatial aggregation scales (see, for example, the applications of GAMs to mid-term forecasting Zimmermann and Ziel 2025 and to local load signals Lambert et al. 2023). These different electricity demand forecasts play a crucial role in the fight against global warming. On the one hand, they facilitate the massive integration of renewable energy (by reducing uncertainty about demand, we can afford to increase uncertainty about electricity production). They also make it possible to adjust the demand of both individual and industrial actors, for example, to run large calculations or start machines at times when demand is low and production is guaranteed by low-carbon emission sources.

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

# A  Detailed structure of generalized additive models (fixed setting)

In the following, we detail the two steps required to set a GAM formula in our set-up. We recall that the target variable is a time series $(Y_t)_{t \geq 1}$ where $Y_t$ represents the amount of electricity demanded between times $t$ and $t + 1$. The latter depends on a multitude of variables $Z_t$, including weather (temperature, wind, etc.) and calendar-related (public holidays, weekdays, etc.) factors. The set of covariates $X_t$ used in the GAM formula includes actual and past covariates, so $X_t = (Z_s)_{s=1,\dots,t}$ and we obtain the equation

$$\mathbb{E}[Y_t | X_t] = \sum_{k=1}^{K} f_k(X_{J_k, t}).$$

The specification of the smooth functions $f_k$ is divided into two: feature engineering (to extract useful information from $X_{J_k, t}$) and relationship specification (to link this information with the target variable).

**Feature engineering.** For any $k = 1, \dots, K$, feature engineering for covariates $X_{J_k}$ may be relevant. We detail in the following the two examples that we consider.

*Example 1: Exponential smoothing of the temperature.*  The incorporation of exponentially smoothed temperatures, which model the thermal inertia of buildings, is likely to improve electricity demand forecasts; see, e.g. Goude et al. (2013). The creation of such a variable is based on a smoothing parameter $\alpha \in [0, 1]$ and requires access to all past temperature values, so we consider, for any time step $t$, $X_{J_k, t} = (T_1, \dots, T_t)^\top$, where $T_t$ denotes the temperature at time step $t$. The exponentially smoothed temperature parameterized by $\alpha$ is defined by:

$$\psi_k(T_t) = \begin{cases} T_1 & \text{if} \quad t = 1 \\ \alpha \psi_k(T_{t-1}) + (1 - \alpha)T_t & \text{else.} \end{cases}$$

By induction, for any time step $t = 1, 2 \dots$, we get $\psi_k(X_{J_k, t}) = \sum_{s=0}^{t-1}(1 - \alpha)\alpha^s T_{t-s} + \alpha^t T_1$. We emphasize that the function $\psi_k$ depends only on the smoothing parameter $\alpha$.

*Example 2: Categorical variable selection.*  We may consider a categorical variable $D$ taking integer values between 1 and $m$, and possibly a default value 0. Among the $m$ modalities, the objective is to select only a subset of them. Let $v$ a vector in $\{0, 1\}^m$ such that $v_j = 1$ if the modality $j$ is selected and 0 otherwise. We define the engineering function $\psi_k$ for the feature $D$ that depends only on $v$ by

$$\psi_k(D_t) = D_t \mathbf{1}_{v_{D_t}=1}.$$

When no feature engineering is required, the function $\psi_k$ is the identity. To consider bi-variate effects, with, e.g. some exponentially smooth temperature and categorical variable selection, the above functions of Examples 1 and 2 can be combined.

**Relationship specification.** Let $\tilde{f}_k$ be the GAM effect associated to the engineered feature $\psi_k(X_{J_k})$, namely the function such that $f_k(X_{J_k}) = \tilde{f}_k(\psi_k(X_{J_k}))$. In the GAM framework, we take $\tilde{f}_k$ in the linear span of a set of $q_k$ basis functions. More precisely, with $b_{k,i}$ the $i^{\text{th}}$ basis function, and any $x$ belonging to set of the values of the covariate $\psi_k(X_{J_k})$, $\tilde{f}_k$ is assumed to have the representation

$$\tilde{f}_k(x) = \sum_{i=1}^{q_k} \beta_{k,i} b_{k,i}(x),$$

for some values of the unknown parameters $\beta_{k,i}$. For a univariate smooth effect, that is, when $\psi_k(X_{J_k})$ is a quantitative real variable, classical basis functions are obtained using cubic splines, cyclic cubic splines, thin plate splines, etc. - see Wood et al. (2015) for further details. We emphasize

that the above notation also works for tensor products, i.e. when $\tilde{f}_k$ takes any number $m$ of covariates as input. By defining a spline basis $(b_i)_{i=1,\dots,q_j}$ for each component $j$ of the already engineered feature vector $\psi_k(X_{J_k})$, with $x_j$ belonging to the set of values of feature $j$, we consider

$$\tilde{f}_k(x_1,\dots,x_m) = \sum_{i_1=1}^{q_1}\sum_{i_2=1}^{q_2}\cdots\sum_{i_m=1}^{q_m} \beta_{i_1,i_2,\dots,i_m} b_{i_1}(x_1)b_{i_2}(x_2)\dots b_{i_m}(x_m) = \sum_{i=1}^{q_1 q_2\dots q_m} \widetilde{\beta}_i \widetilde{b}_i(x_1, x_2,\dots,x_m),$$

where $\widetilde{\beta}_i = \beta_{i_1,i_2,\dots,i_m}$ and $\widetilde{b}_i(x_1, x_2,\dots,x_m) = b_{i_1}(x_1)b_{i_2}(x_2)\dots b_{i_m}(x_m)$ with, for $l = 1,\dots,m$,

$$i_l = \left\lceil \frac{r(i, q_1 q_2 \dots q_{l-1}) - 1}{q_{l+1}q_{l+2}\cdots q_m} \right\rceil + 1\,, \text{ where } r(i, q) \text{ is the remainder of the Euclidean division of } i \text{ by } q\,.$$

Finally, when $\psi_k(X_{J_k})$ is a categorical variable, taking integer values between 1 and $m$, we set $q_k = m$ and $b_i(x) = \mathbf{1}_{x=i}$, for $i = 1,\dots,m$. Thus, a coefficient $\beta_{k,i}$ is estimated of each modality $i$.

## B  Algorithms

We aim to find the best adaptive Generalized Additive Model (GAM) among a search-space $\Omega$ defined in Section 3. We recall that we used the P-IRLS algorithm from the `mgcv` R-package to train the GAMs. In what follows, for a formula $f$ and a training data set $\mathcal{D}_{\text{TRAIN}}$, we denote by $\widehat{f} = \text{P-IRLS}(f, \mathcal{D}_{\text{TRAIN}})$ the trained GAM. In Chapter 5 of de Vilmarest (2022), Algorithm 5 proposes an iterative grid search procedure to optimize the matrix which parameterizes the adaptive version of a model $\widehat{f}$. In the following, we denote by $\text{Q}_{\text{IGS}}(\widehat{f}, \mathcal{D}_{\text{TRAIN}}, Q_0)$ the matrix output of this algorithm, for a model $\widehat{f}$ and an initial matrix $Q_0$. Once the model $(f, Q) \in \Omega$ has been trained, it is possible to evaluate its performance. To do so, we use the loss function defined in Equation (2) which takes as arguments the formula $f$, the hyperparameter $Q$ and the data set $\mathcal{D}$. As detailed in Section 4, $\ell(f, Q, \mathcal{D})$ requires the calculation of forecasts in the adaptive setting. In what follows, if $Q = -\infty$, the forecasts are computed in the fixed setting (namely with $\widehat{\theta}_t = \mathbf{1}$ for all $t$). Table 2 contains the notations used in the algorithms defined below. Algorithm 1 describes the generation of a model $(f, Q) \in \Omega$. Algorithm 2 is declined in two versions: $\text{EA}(f) + \text{Q}_{IGD}$ and $\text{EA}(f, Q)$. It explores $T$ models by evolving a population of constant size $M$. To do so, it starts with $M$ sampled models (using Algorithm 1). At each round, it creates two new models from two good ones using evolutionary operators and then deletes the two models with the highest loss among the entire population (including the two new ones). In the first version $\text{EA}(f) + \text{Q}_{IGD}$, the evolutionary algorithm is used only to optimize the GAM formula, and in a second step, the hyperparameter for its adaptive version is optimized using the iterative grid-search $\text{Q}_{IGD}$. In contrast, in $\text{EA}(f, Q)$, we do not use the iterative grid search and the evolutionary algorithm optimizes the formula and the hyperparameter of the adaptive version both at the same time.

## C  Experiments: state-of-the-art handcrafted model

We refer, among others, to Goude et al., 2013 and Gaillard et al., 2016 for an exhaustive presentation of the generalized additive models used to forecast power demand. Our state-of-the-art handcrafted model takes into account some meteorological variables at an hourly time step: the temperature $T_t$ and the smoothed temperature $\bar{T}_t$, the cloud cover $C_t$, and the wind speed $W_t$; and some calendar variables: the day of the week $D_t$, the hour of the day $H_t \in \{1, \dots, 24\}$ and the position in the year $P_t \in [0, 1]$, which takes linear values between $P_t = 0$ on January 1st at 00:00 and $P_t = 1$ on December the 31st at 23:59. We also add a "Break" variable $B_t$: a categorical variable refering to various moment in the year (Winter time, Summer time, August and Christmas break, etc.). The

| | |
|---|---|
| $t$ | Time step within the time series |
| $i$ | Iteration of a search algorithm |
| $f$ | Formula of a Generalize Additive Model (GAM) |
| $Q$ | Hyperparameter of the adaptive version of a GAM ($K \times K$-matrix, where $K$ is the number of effects of the GAM) |
| $\Omega$ | Search space containing the GAM formulae and the associated matrices |
| $\mathcal{D}_{\text{TRAIN}}$ | Data set used for model training (P-IRLS and $Q_{\text{IGS}}$ algorithms) |
| $\mathcal{D}_{\text{VALID}}$ | Data set used for model evaluation |
| $\mathcal{D}_{\text{TEST}}$ | Data set used for AutoML algorithms' performance evaluation |
| | |
| | Algorithms and operators |
| | |
| $\mathcal{B}(p)$ | Function to generate a realization of a random variable that follows a Bernoulli distribution of parameter $p$ |
| P-IRLS$(f, \mathcal{D}_{\text{TRAIN}})$ | GAM output by P-IRLS algorithm (`mgcv` R-package) for a formula $f$ and a training data set $\mathcal{D}_{\text{TRAIN}}$ |
| $\ell(f, Q, \mathcal{D}_{\text{VALID}})$ | Loss defined in Equation (2) |
| $Q_{\text{IGS}}(\widehat{f}, \mathcal{D}_{\text{TRAIN}}, Q_0)$ | Adaptive version hyperparameter for trained model $\widehat{f}$ output by Algorithm 5 of de Vilmarest (2022) for an initial hyperparameter $Q_0$ and a training data set $\mathcal{D}_{\text{TRAIN}}$ |
| $\mathcal{M}(f, Q)$ | Mutant model created from $(f, Q) \in \Omega$ using the mutation operator described in Section 5 |
| $\mathcal{CO}\big((f_1, Q_1), (f_2, Q_2)\big)$ | Children models created from models $(f_1, Q_1)$ and $(f_2, Q_2) \in \Omega$ using the crossing-over operator described in Section 5 |

Table 2: Table of Notation

state-of-the-art handcrafted model is then the sum of 24 daily models, one for each hour of the day. Thus, at any time step $t$, with $h = H_t$, the following model is considered:

$$\widehat{f}^h(X_t) = \widehat{s}_T^h(T_t^h) + \widehat{s}_{\overline{T}}^h(\overline{T}_t^h) + \widehat{s}_C^h(C_t^h) + \widehat{s}_W^h(W_t^h) + \sum_{d=1}^{D} \widehat{\delta}_d^h \mathbf{1}_{D_t=d} + \sum_{b=1}^{B} \widehat{\beta}_b^h \mathbf{1}_{B_t=b} + \widehat{s}_P^h(P_t^h).$$

The $\widehat{s}_T^h$, $\widehat{s}_{\overline{T}}^h$, $\widehat{s}_C^h$, $\widehat{s}_W^h$, and $\widehat{s}_P^h$ functions catch the effect of the the meteorological variables and of the yearly seasonality. They are cubic splines: $\mathcal{C}^2$-smooth functions made up of sections of cubic polynomials joined together at points of a grid. The coefficients $\widehat{\delta}_d^h$ and $\widehat{\beta}_b^h$ model the influence of the day of the week and the period of the year respectively; these effects are represented as a sum of indicator functions. As we consider a model per hour, all the coefficients and splines are indexed by $h$. In our automated model selection approach, we consider other variables such as the month of the year, the maximum and minimum temperatures per day, the binary variable Weekend, and so on.

**Algorithm 1** Generation of an (adaptive) GAM model

---

**Inputs**:

    Covariate list $L_{\text{VAR}}$

    Categorical covariates list $L_{\text{CAT\_VAR}} \subset L_{\text{VAR}}$

    Spline list $L_{\text{SP}}$ (for the uni-variate effects), Tensor list $L_{\text{TE}}$ (for the bi-variate effects)

    Maximum number of effects in the GAM $K_{\text{max}}$. (If None, use the number of features),

    Minimum / Maximum degree of freedom for a spline $k_{\text{min}}$ / $k_{\text{max}}$,

    Probability of generating a bi-variate effect $p_{\text{bi\_var}}$,

    Minimum / Maximum smoothing coefficient for the temperature $\alpha_{\text{min}}$ / $\alpha_{\text{max}}$,

    List of days / offsets to be calculated $L_{\text{DAY}}$ / $L_{\text{OS}}$,

    Boolean specifying whether $Q$ should also be generated or not Kalman

    Minimum / Maximum value for the $Q$ matrix coefficients $Q_{\text{min}}$ / $Q_{\text{max}}$,

    Minimum / Maximum value for the $Q$ matrix coefficients' standard deviation $\sigma_{\text{min}}$ / $\sigma_{\text{max}}$

**Initialization**

    Sample a number of effects $K \in \{1, \ldots, K_{\text{max}}\}$

    Create an empty list $L_{\text{GAM}}$ that will contain the effects of the generated GAM

    **If** Kalman = True:

        Uniformly choose a standard deviation $\sigma$ between $\sigma_{\text{min}}$ and $\sigma_{\text{max}}$,

        Logarithmically choose the $Q$ matrix's diagonal coefficients between $Q_{\text{min}}$ and $Q_{\text{max}}$,

        Append a list containing $\sigma$ and the diagonal coefficients to $L_{\text{GAM}}$

    **For** $k = 1, \ldots, K$:

        Sample $X_k \sim \mathcal{B}(p_{\text{bi\_var}})$

        **If** $X_t = 1$, a bi-variate is generated:

            Select two distinct covariates from $L_{\text{VAR}}$ while making sure they do not already exist as a bi-variate effect in $L_{\text{GAM}}$ and that they are not both categorical using $L_{\text{CAT\_VAR}}$,

            Sample a relationship in $L_{\text{SP}}$ if one of the covariates is in $L_{\text{CAT\_VAR}}$ and a degree of freedom between $k_{\text{min}}$ and $k_{\text{max}}$; or in $L_{\text{TE}}$ and two degrees of freedom otherwise while ensuring compatibility with the chosen relationship

        **Else**:

            Select a covariate while making sure it does not already exist in $L_{\text{GAM}}$

            Sample a relationship in $L_{\text{SP}}$ and a degree of freedom while ensuring compatibility with the chosen relationship

        If the covariate(s) are temperature, days or offsets, generate a temperature smoothing coefficient between $\alpha_{\text{min}}$ and $\alpha_{\text{max}}$, or a list among $L_{\text{DAY}}$ or $L_{\text{OS}}$

        Add the effect to $L_{\text{GAM}}$

    **Output**: $L_{\text{GAM}}$

---

---

**Algorithm 2** Steady state evolutionary algorithm (EA) for adaptive GAM selection

---

**Inputs:**

Training data set $\mathcal{D}_{\text{TRAIN}}$, validation data set $\mathcal{D}_{\text{VALID}}$, loss function $\ell$

P-IRLS algorithm, Mutation operator $\mathcal{M}$, Crossover operator $\mathcal{CO}$,

Population size $M$, Size of the tournament selection $m \in \{1, \ldots, M-1\}$,

Parameter for the initialization of the iterative grid search $q_0$, Number of tested models $T$,

$V_{\text{EA}} \in \{\text{EA}(f) + Q_{IGD}, \text{EA}(f, Q)\}$ the version of EA considered

**Initialization**

For $i = 1, 2, \ldots, M$:

   **If** $V_{\text{EA}} = \text{EA}(f, Q)$:

      Sample a model $(f_i, Q_i)$ using Algorithm 1 with KALMAN = TRUE

      Train the model: $\widehat{f_i} = \text{P-IRLS}(f_i, \mathcal{D}_{\text{TRAIN}})$

   **Else**:

      Sample a formula $f_i$ using Algorithm 1 with KALMAN = FALSE

      Train the associated fixed model: $\widehat{f_i} = \text{P-IRLS}(f_i, \mathcal{D}_{\text{TRAIN}})$

      $Q_i = -\infty$ (when $V_{\text{EA}} = \text{EA}(f) + Q_{IGD}$)

   Get the loss $\ell_i = \ell\big(f_i, Q_i, \mathcal{D}_{\text{VALID}}\big)$

For $i = 1, \ldots, \lfloor \frac{T-M}{2} \rfloor$:

Sample a partition $\mathcal{I} \in \{1, \ldots, M\}$ of size $m$ and select $j_A \in \text{argmin}_{j \in \mathcal{I}} \ell_j$

Sample a partition $\mathcal{J} \in \{1, \ldots, M\} \backslash \{j_A\}$ of size $m$ and select $j_B \in \text{argmin}_{j \in \mathcal{J}} \ell_j$

Apply the crossover operator on models $j_A$ and $j_B$:

   $\big((f^{\text{NEW1}}, Q^{\text{NEW1}}), (f^{\text{NEW2}}, Q^{\text{NEW2}})\big) = \mathcal{CO}\big((f_{j_A}, Q_{j_A}), (f_{j_B}, Q_{j_B})\big)$

Mutate the two new models:

   $(f^{\text{NEW1}}, Q^{\text{NEW1}}) = \mathcal{M}\big(f^{\text{NEW1}}, Q^{\text{NEW1}}\big)$ and $(f^{\text{NEW2}}, Q^{\text{NEW2}}) = \mathcal{M}\big(f^{\text{NEW2}}, Q^{\text{NEW2}}\big)$

Train two the new models: $\widehat{f}^{\text{NEW1}} = \text{P-IRLS}(f^{\text{NEW1}}, \mathcal{D}_{\text{TRAIN}})$, $\widehat{f}^{\text{NEW2}} = \text{P-IRLS}(f^{\text{NEW2}}, \mathcal{D}_{\text{TRAIN}})$

**If** $V_{\text{EA}} = \text{EA}(f, Q)$:

   $Q^{\text{NEW1}} = Q_{IGD}(\widehat{f}^{\text{NEW1}}, \mathcal{D}_{\text{TRAIN}}, Q^{\text{NEW1}}, N)$ and $Q^{\text{NEW2}} = Q_{IGD}(\widehat{f}^{\text{NEW2}}, \mathcal{D}_{\text{TRAIN}}, Q^{\text{NEW2}}, N)$

Get the losses: $\ell^{\text{NEW1}} = \ell\big(f^{\text{NEW1}}, Q^{\text{NEW1}}, \mathcal{D}_{\text{VALID}}\big)$, $\ell^{\text{NEW2}} = \ell\big(f^{\text{NEW2}}, Q^{\text{NEW2}}, \mathcal{D}_{\text{VALID}}\big)$

**If** $\ell^{\text{NEW1}} < \max_{j \in \{1, \ldots, M\}} \ell_j$:

   Replace $(f_{j_{\text{OLD}}}, Q_{j_{\text{OLD}}})$ where $j_{\text{OLD}} \in \text{argmax}_{j \in \{1, \ldots, M\}} \ell_j$ by $(f^{\text{NEW1}}, Q^{\text{NEW1}})$

**If** $\ell^{\text{NEW2}} < \max_{j \in \{1, \ldots, M\}} \ell_j$:

   Replace $(f_{j_{\text{OLD}}}, Q_{j_{\text{OLD}}})$ where $j_{\text{OLD}} \in \text{argmax}_{j \in \{1, \ldots, M\}} \ell_j$ by $(f^{\text{NEW2}}, Q^{\text{NEW2}})$

Select the best model $(f_{j_{\text{BEST}}}, Q_{j_{\text{BEST}}})$ with $j_{\text{BEST}} \in \text{argmin}_{j=1, \ldots, M} \ell_j$

**If** $V_{\text{EA}} = \text{EA}(f) + Q_{IGD}$ :

   $Q_{j_{\text{BEST}}} = Q_{IGD}(\widehat{f}_{j_{\text{BEST}}}, \mathcal{D}_{\text{TRAIN}}, Q_0)$ with $Q_0 = q_0 I_d$

**Output:** $(f_{\text{BEST}}, Q_{\text{BEST}})$

---

