# OpenReview forum: "AutoML Algorithms for Online Generalized Additive Model Selection: Application to Electricity Demand Forecasting"
_automl.cc/AutoML/2025/ABCD_Track — AutoML 2025 ABCD Track_

### Review · Reproducibility_Reviewer_hdqf · 2025-04-28

**Comments To Authors:**

Thank you to the authors for the `README` and the clarity of the reproduction instructions. Due to limited resources, I only ran the code for one sample hour. Yet, the environment setup and running of the commands did not cause an obstacle, and following the provided notebook was seamless.

I would generally recommend providing some expected results for the smaller example cases for the cases of limited compute as in my situation, to fully validate the results.

Given the ease of setting up the environment and successfully running the intended code paths, I recommend **acceptance** of this work from a reproducibility standpoint.

**Review Confidence:**

3

**Review Rating:**

8

---

### Official Review · Reviewer_ByLZ · 2025-04-30

**Comments To Authors:**

Summary Of Contributions:
The authors propose optimizing the hyperparameters of GAM using the DRAGON package of Keisler (2025), originally designed for neural architecture search. They propose that the work generalizes it for automated online generalized additive model selection by defining an efficient modeling of the search space (namely, the space of the GAM formulae and adaptation parameters).

Potential Impact On The Field:
The paper does not propose the impact and novelty of the idea/application well. Therefore, I would not say there are potential considerable impact on the field for the work.

Technical Quality, Correctness, Clarity:
The technical quality of the work is weak even though the work seems correct. In terms of clarity, the paper needs to be improves. I suggested some of the improvements below.

Suggestions:
1. The introduction should be well-structured, and I would recommend rewriting in order. There are no paragraphs and it is not clear.
2. I would recommend that the Abstract should not contain any citations. It decreases the paper quality.
3. The introduction contains some colourful optimisation problem definitions and refers to sections in the paper. It does not look professional as the fonts are different, and the equations do not look correct.
4. Literature review should be comprehensive as there are enough page limitation for such chapter if the authors stick on specific literature and application.


Ethics And Accessibility:
There are no ethical consideration.

Overall Review:
Reject

**Review Confidence:**

5

**Review Rating:**

3

---

### Official Review · Reviewer_UXe2 · 2025-04-30

**Comments To Authors:**

The work proposes an application of AutoML algorithms for the optimization of GAMs in the domain of electricity demand forecasting. To this end, the work proposes to use the DRAGON library for this purpose and discusses required modifications to be able to apply this AutoML tool in the desired domain. This setup aligns well with the requirements for AutoML applications of the ABCD track.
Further, the work shows the utility of the proposed approach by evaluating the proposed algorithms for forecasting of the hourly french electricity demand. The proposed tool achieves the lowest RMSE and readily outperforms a handcrafted baseline which is considered SOTA by the authors.

While I am not very familiar with the related literature, the work was mostly easy to follow and provided enough details to highlight why certain design decisions where made or needed. Still, in the beginning I had difficulty to see the connection to AutoML beyond the use of the DRAGON package. But that is due to an initial misunderstanding on my part on the role of the P-IRLS algorithm in the setup and only became clearer after re-reading parts of the work.

Overall I believe that the work proposes an interesting and important application that could facilitate active discussion at the conference and could lead to even more work on using AutoML for tuning models of energy demand forecasting.

**Review Confidence:**

2

**Review Rating:**

8

---

### Official Review · Reviewer_K2Di · 2025-05-01

**Comments To Authors:**

This is a nice paper! A clear extension of existing frameworks to handle AutoML of additive models and applied to a real-world problem setting. The paper is clearly written and well-presented.

Specific comments:

- Are there any bounds/range on valid values for Q and how is this handled? I couldn't find this in the paper.
- The regularization parameter, eta would seem to be potentially important and the results sensitive to it. Perhaps you could comment on that? Where does the number "5000" come from in the setting of this parameter?

**Review Confidence:**

4

**Review Rating:**

8

---

### Official Review · Reviewer_WWoc · 2025-05-06

**Comments To Authors:**

The methodology is sound and well-motivated. The authors correctly identify that both the GAM formula (choice of effects and bases) and the adaptation hyperparameter (state-space noise covariance Q) critically affect forecast accuracy. Automating these choices with AutoML is a suitable strategy given the combinatorial search space. They proposed optimizing the hyperparameters using the DRAGON package, originally for neural architecture search. Extending DRAGON to GAM model selection is an innovative idea.

The encoding of hyperparameters is comprehensive, covering univariate/bivariate effects, smoothed features, and multiple spline bases.

I have the following concerns:

1. The proposed AutoML process is computationally heavy. The paper mentions limiting formula complexity but gives no runtime analysis.

2. The authors noted the need for cross-validation. As is, it is hard to assess generalizability. Additional validation (e.g., multiple runs, bootstrap) would strengthen confidence.

3. The method considers only diagonal matrices for $Q$. Would it restrict the performance of the adaptive model, compared to optimizing a general positive-semidefinite matrix? Also, please clarify if the current encoding can be extended to represent a general positive-semidefinite matrix.

4. In Eq. (2), $DF(f)$ is used as the regularization term. Would it make more sense to consider AIC, BIC, or L^2 norm of $f$ as regularization terms?

5. How sensitive are results to the EA’s randomness (initial population, crossover/mutation outcomes)? The paper does not report confidence intervals or multiple trials.

6. The method fixes the Kalman filter’s initial state and only learns Q. How were initial values, e.g., initial covariance, chosen? Could learning these (or using a data-driven initialization) improve adaptability?

7. The paper only experiments with an EA. Since DRAGON supports other algorithms (e.g., Hyperband), have the authors tried those? For example, hyperparameter tuning via Bayesian optimization might find good formulas more quickly.

**Review Confidence:**

4

**Review Rating:**

9

---

### Meta-Review · Area_Chair_fpqe · 2025-05-09

**Recommendation:** Accept
**Confidence:** 3

**Metareview:**

The paper discusses the work involved in applying an AutoML tool (Dragon package) to electricity demand forecasting. They show the impact on tuning the hyperparameters of a GAM model (required for interpretability for the use-case) against a hand-crafted previous method.

The reviewers highlighted the relevance of the application and the value to present this impactful application to the AutoML community as a potential real-world use-case and mostly voted for acceptance while the reviewer that voted for reject raised points that mostly concern formatting and could be potentially adressed by the author easily. As a result, I recommend to accept the paper.